# An international validation of the Bolton Unistride Scale (BUSS) of tenacity

**Chathurika Kannangara**[1]*, **Rosie Allen**[1]*, **Kevin D. Hochard**[2], **Jerome Carson**[1]

**1** University of Bolton, Bolton, Greater Manchester, England, United Kingdom, **2** University of Chester, Chester, England, United Kingdom

* c.kannangara@bolton.ac.uk (CK); rea1eps@bolton.ac.uk (RA)

## Abstract

Academic success at University is increasingly believed to be a combination of personal characteristics like grit, resilience, strength-use, self-control, mind-set and wellbeing. The authors have developed a short 12-item measure of tenacity, the Bolton Uni-Stride Scale (BUSS) which incorporates these elements. Previous work in the UK had established the reliability and validity of the BUSS. The present paper reports the findings of an International validation of BUSS across 30 countries (*n* = 1043). Participants completed the BUSS alongside other recognised scales. Factor analysis revealed an almost identical two-factor solution to previous work and the reliability and validity of the scale were supported using an international sample. The authors recommend however that the scale be used as a single score combining all 12 items. In the light of this, the authors suggest that the BUSS will be a useful measure to incorporate in studies of academic attainment.

**Data Availability Statement:** All relevant data are within the paper and its Supporting information files (An International Validation of BUSS Dataset. sav).

## Introduction

### Tenacity

There is now accumulating empirical evidence to show that academic success at university is reliant on a combination of personal characteristics. Academic success cannot simply be attributed to any one quality, but is shaped through a number of characteristics such as grit, resilience, strengths-use, self-control, mind-set and well-being [1]. The complex and multidisciplinary nature of these characteristics work together to give university students the best opportunity for educational success. Ultimately, it is the combination of these characteristics that equate to how tenacious a student will be in their academic studies. Together, these characteristics promote tenacity and success in university and arguably in later life. 'Tenacity' can be defined as the combination of grit, resilience, strengths-use, self-control, psychological well-being and a growth mind-set, that provides students with the capacity to thrive at university [1]. A tenacious student expresses a passion and perseverance for pursuing long-term goals [2] and has the ability to overcome adversity and continue striving despite challenges they may face [3]. A student who is particularly tenacious in their academic studies will use their personal strengths in different situations and work towards improving on their strengths [4,5]. They can regulate their study habits, control their impulses and manage their own

**Funding:** CK was awarded a small grant of £2000 from the University of Bolton to pay for the BUSS survey on the Prolific website. The funders had no role in study design, data collection and analysis, decision to publish, or preparation of the manuscript.

**Competing interests:** The authors have declared that no competing interests exist.

actions and behaviors, such as switching their smartphone off to avoid distraction [6,7]. Adopting a growth mind-set, they also believe that their efforts can improve their abilities and perceive setbacks as a springboard to success [8]. Finally, tenacious students frequently experience positive affect and possess better mental health outcomes [9].

## Worldwide university students

Research from different countries around the world has shown that these characteristics are universal and play a crucial role with university students internationally. For instance, empirical evidence has shown that grit is an essential personal resource among university students, with similar findings from various countries including Germany [10], Spain [11], South Africa [12], Australia [13], China [14], Finland [15] and the USA [16]. Likewise, resilience is considered to be a universal concept that is necessary for various academic and mental health outcomes in university students, such as academic performance [17,18], motivation and effort [19], academic achievement [20,21] and general well-being [22]. Again, these findings are shared across many countries, including the USA, Kenya, Spain, Iran and the Czech Republic, to name but a few. Clearly, these characteristics are universal and strongly influence the educational success and well-being of university students. Therefore, tenacity is proposed to be a universal concept and so the need for a measure that is applicable to university students across the world would be beneficial.

## Present study

There is a need for a reliable and valid measure of these collective characteristics in a short and concise tool, to improve the quality of research in this area. The Bolton Uni-Stride Scale (BUSS) was specifically devised to capture students' tenacity. There was good support for the reliability and validity of the BUSS [1], in a series of UK based studies. Originally, the BUSS was developed based on British samples of university students, and to date, has not been further validated with other populations. This paper reports an international validation study of the BUSS, on a large sample of university students from across the world. This will allow for the development of an international tool for the measurement of tenacity in university students.

## Materials and methods

A web-based survey was designed to facilitate data collection from university students on an international scale. Previous reliability and validity testing of the BUSS was carried out on a sample of university students from the United Kingdom. This research aimed to test whether the factor structure of the BUSS is supported using an international sample and to examine other aspects of the reliability and validity of the BUSS. The present study used the Prolific website to recruit participants which enabled representation from a range of countries. Using Prolific also meant that the researchers could pre-screen participants as per the study requirements [23], which was useful for recruiting participants who matched specific criteria. Although participants are from different countries around the world, it was made sure, using selection criteria in Prolific, that all participants' primary language was English. Therefore no adaptations of the BUSS were needed to account for language differences.

### Participants

A total of 1043 participants was recruited (*See* Table 1 *for the full description of participants*).

**Table 1. Demographic characteristics of the participant sample.**

| Demographic Characteristic | | Number of Participants (N) | Percentage of Sample (%) |
|---|---|---|---|
| Age | 16–18 | 22 | 2.1 |
| | 18–25 | 720 | 69.0 |
| | 25–30 | 179 | 17.2 |
| | 30+ | 122 | 11.7 |
| Education Level | College | 179 | 17.2 |
| | University–First Year | 161 | 15.4 |
| | University–Second Year | 151 | 14.5 |
| | University–Third Year | 320 | 30.7 |
| | Postgraduate–Masters | 197 | 18.9 |
| | PhD/Doctoral Studies | 35 | 3.4 |
| Country of Residence | Australia | 15 | 1.4 |
| | Austria | 3 | .3 |
| | Belgium | 7 | .7 |
| | Canada | 67 | 6.4 |
| | Czech Republic | 4 | .4 |
| | Denmark | 1 | .1 |
| | Finland | 2 | .2 |
| | France | 6 | .6 |
| | Germany | 49 | 4.7 |
| | Greece | 41 | 3.9 |
| | Hungary | 12 | 1.2 |
| | Iceland | 1 | 1.1 |
| | India | 2 | .2 |
| | Ireland | 6 | .6 |
| | Israel | 2 | .2 |
| | Italy | 40 | 3.8 |
| | Malaysia | 1 | .1 |
| | Mexico | 68 | 6.5 |
| | Netherlands | 13 | 1.2 |
| | New Zealand | 3 | .3 |
| | Poland | 102 | 9.8 |
| | Portugal | 119 | 11.4 |
| | Slovenia | 5 | .5 |
| | Spain | 29 | 2.8 |
| | Sweden | 3 | .3 |
| | Switzerland | 2 | .2 |
| | Turkey | 1 | .1 |
| | United Kingdom | 38 | 3.6 |
| | United States of America | 399 | 38.3 |
| | Other/Not Specified | 2 | .2 |

## Measures

**Bolton Uni-Stride Scale (BUSS).** BUSS is a short and concise measure of tenacity that was developed to incorporate important characteristics such as grit, resilience, self-control and well-being [1]. This twelve-item scale measures persistence and self-composure. Persistence is measured through seven items and Self-composure is measured through five items. A sample item from Persistence is *"I consider myself as very capable in handling personal challenges"*. All

**Table 2. Correlation between each item of BUSS and total BUSS score from several studies.**

| BUSS items | Sample 1 | Sample 2 | Sample 3 | Sample 4 | Current Sample (International) |
|---|---|---|---|---|---|
| | N = 1087 | N = 933 | N = 331 | N = 146 | N = 1043 |
| 1 | .518 | .578 | .531 | .610 | **.620** |
| 2 | .540 | .437 | .493 | .545 | **.481** |
| 3 | .577 | .647 | .550 | .551 | **.641** |
| 4 | .512 | .502 | .490 | .473 | **.530** |
| 5 | .523 | .439 | .517 | .637 | **.426** |
| 6 | .623 | .630 | .618 | .727 | **.663** |
| 7 | .538 | .560 | .621 | .713 | **.676** |
| 8 | .242 | .006 | .259 | .413 | **.197** |
| 9 | .546 | .503 | .505 | .642 | **.522** |
| 10 | .555 | .497 | .516 | .541 | **.544** |
| 11 | .595 | .612 | .613 | .595 | **.616** |
| 12 | .563 | .579 | .520 | .570 | **.598** |

Notes: Sample 1 was 1087 students from a university in the North West of England [25] Sample 2 was 933 adolescents from the North West of England [26]. Sample 3 was 331 students from a university in the North West of England [25]. Sample 4 was 146 students from a university in the North West of England [27].

items from Persistence are positively keyed. These items are scored on a Likert-type scale (*1 = Strongly Disagree to 5 = Strongly Agree*). A sample item from Self-Composure is *"I do things that feel good in the moment, but later regret"*. All items from Self-Composure are negatively keyed. These items are scored on a Likert-type scale (*1 = Strongly Agree to 5 = Strongly Disagree*). As recommended, this scale consists of positively and negatively phrased items to minimise desirability bias [24], whereby negatively phrased items were reverse coded prior to analysis to ensure all items were weighted equally. Table 2 demonstrates that all items of the BUSS correlate significantly with the total BUSS score. With the exception of item 8, all correlate more than .20, but no more than .80. On inspection of Table 2, the correlations among the twelve items of BUSS and total BUSS remain relatively consistent throughout different studies using independent samples. Table 2 also shows that this is confirmed, when utilising an international sample of university students. Previous research confirmed the reliability and validity of BUSS as internal consistency reliability was good (.74) and three week test-retest reliability was good (>.70). Also, discriminant and convergent validity were good [1].

**Self-control.** The Self-Control Scale (SCS) is a ten-item self-control scale [28]. This scale is scored in the form of a rating scale with participants responding to ten statements from "not at all like me" (1) to "very much like me" (10). The scale included statements like *"I'm good at resisting temptation"*. Test-retest reliability was high at .89 and internal consistency estimates of reliability were also high, showing a Cronbach's alpha estimate of .89 [28]. Similar internal consistency estimates were reported using the participant sample from the present study, showing a Cronbach's alpha estimate of .83. This confirms that the scale appears to have good internal consistency.

**Future work self.** Participant's future work self was measured by the Future Work Self Scale, developed by [29]. This is a 5-item scale that uses a 5-point Likert response format from 1 ("Strongly Disagree") to 5 ("Strongly Agree"). Items in the scale included *"I am very clear about who and what I want to become in my future work"*. This scale has shown good construct validity, predictive validity and internal consistency reliability [29]. Internal consistency estimates using the concrete data were good, showing a Cronbach's alpha estimate of .93.

**Resilience.** The Connor-Davidson Resilience Scale (CD-RISC 10) is a ten-item resilience scale that is scored in the form of a rating scale–from (0) "not true at all" to (4) "true nearly all the time" [30]. For instance, statements included *"I am not easily discouraged by failure"*. This scale demonstrates good test-retest reliability (.90) over a two week period [31]. Reliability analysis shows good internal consistency [31], which was mirrored in the present study which found a Cronbach's alpha estimate of .87.

**Mental well-being.** The mental well-being of participants was measured using the Warwick-Edinburgh Mental Wellbeing Scale (WEMWBS). This is a 14-item scale that focuses on feelings and functioning related to mental well-being. The WEMWBS is scored on a 5-point Likert scale from "None of the time" to "All of the time". All 14 items in this scale are positively phrased. On example item states *"I've been dealing with problems well."* One-week test-retest reliability was high on a sample of British university students [32]. WEMWBS also showed good convergent validity [32], construct validity [33], concurrent validity [32] and discriminant validity [32]. This scale has shown high internal consistency in a sample of UK university students [32]. Internal consistency estimates from the present study were good, showing a Cronbach's alpha estimate of .91.

**Grit.** This study used the Short Grit Scale, developed by Duckworth & Quinn in 2009 [34]. The Grit-S is an 8-item measure of perseverance and passion to pursue long-term goals. This scale includes positively and negatively phrased items, with items 1, 3, 5 and 6 being reverse coded prior to analysis. For instance, one item reads *"I finish whatever I begin"*. The Grit-S is scored in the form of a 5-point Likert scale with anchors of "Very much like me" and "Not like me at all". There is strong evidence for the reliability and validity of this scale. Such that, it has high internal consistency, good test-retest reliability and good predictive validity [34]. High internal consistency estimates were also reported using the participant sample from the present study, showing a Cronbach's alpha estimate of .83.

## Procedure

This study was uploaded to a website called Prolific. Prolific is an online platform for recruiting study participants, where researchers pay individuals to complete their questionnaires [23]. First, participants were asked to read an Information Sheet that described the study and detailed what their involvement will consist of. Participants were made aware that their participation in the study was voluntary. If participants were willing to take part in the study, they were asked to give their consent. Written consent was obtained by asking participants to select "yes" or "no" to the following statement: "*If you would like to participate in the study please consent to take part. If you are not happy to continue with the study, you can withdraw at this point by closing the survey page.*" They were then asked to provide basic demographic information such as age. Following this, they were asked to complete a series of questionnaires that included: BUSS, SCS, FWS, CD-RISC, WEMWBS and Grit-S. Participants were then thanked for their participation. Ethical approval for the study was obtained from the Ethics Committee of the Psychology Department at the University of Bolton in line with British Psychological Society guidelines [35]. The use of Prolific also allows you to check the quality of the responses and data. In this study, one additional items was included in the online survey acting as an attention checker that asked participants "It is important that you pay attention. Please select Strongly Agree". Those participants that failed the attention check question were rejected. It also allows you to access the time it takes participants to complete the survey. For instance, the average time it took a student to complete the online survey was 8.89 minutes. If participants had completed the survey "too quickly", that is they were a statistical outlier and were 3 standard deviations below the mean completion time, they were also rejected.

## Results

In order to confirm the previously established internal structure of the BUSS and explore it's applicability to an international university student sample, a further Confirmatory Factor Analysis (CFA) was conducted.

### Confirmatory factor analysis

The 12 items of the Bolton Uni-Stride Scale (BUSS) were subjected to CFA using SPSS version 23 and AMOS version 26. A maximum likelihood method of factor extraction was deemed most suitable.

Table 3 illustrates the standardised factor loadings from the CFA. Findings demonstrate two latent variables with loadings of >.40 highlighted in bold [36]. As recommended, all of the items loaded strongly on each factor as factor loadings are above 0.4 [36] except for item 5 (.251) and 8 (.199) which loaded weakly onto factor 2 (self-composure).

While all indicator variables load significantly on each latent factor ($p < .01$), on inspection of the Cronbach's alpha estimates, the removal of item 8 led to a small improvement in Cronbach's alpha. Indeed, the Cronbach's alpha of BUSS was .77, while McDonald's omega was .79, which increased to .80 and .81 respectively if item 8 was deleted. Also, the corrected item-total correlation for item 8 was low (.032). Combined, these results indicated that item 8 should be removed from the factor structure. Therefore, item 8 was retained as a contributing item towards BUSS scores overall, however it was removed from the factor structure and further tests of factor analysis and model fit. Item 5 also demonstrated a weak loading onto the self-composure factor (.251) and is below the recommended loading for each item. While there is a very slight increase in Cronbach's alpha, from .799 to .803, and omega, from .814 to .818, if item 5 is removed, this does not appear considerable. It is common practice to remove any item that results in an improvement to the internal consistency [37]. However, the corrected item-total correlation for item 5 indicates good discrimination at .272 and as the 11 items of BUSS demonstrate good reliability at above .70 [38], item 5 was retained.

**Table 3. Maximum likelihood estimates of factor loadings for the BUSS for 1-factor, 2-factor and bifactor CFA.**

| Items of the BUSS(BUSS) | 1-Factor | 2-Factor | | Bifactor | | |
|---|---|---|---|---|---|---|
| | | Persistence | Self-Composure | Persistence | Self-Composure | General |
| **Factor 1 (Persistence)** | | | | | | |
| (7) using personal strengths is a regular habit | **0.771** | .785 | | 0.399 | | **0.882** |
| (1) use my strengths in various situations | **0.706** | .716 | | 0.441 | | **0.819** |
| (6) capable in handling personal challenges | **0.694** | .693 | | -0.071 | | **0.706** |
| (11) generally able to move forward in life | **0.642** | .643 | | 0.004 | | **0.656** |
| (12) always looking for ways to improve talents and skills | **0.639** | .640 | | 0.179 | | **0.668** |
| (3) persistent and hard working | **0.630** | .622 | | 0.185 | | **0.653** |
| (9) find it easy to make decisions | **0.441** | .434 | | -0.233 | | **0.416** |
| **Factor 2 (Self-Composure)** | | | | | | |
| (4) I cannot stop my actions | **0.362** | | .748 | | **0.655** | 0.396 |
| (2) do things that feel good in the moment but later regret | **0.282** | | .636 | | **0.638** | 0.344 |
| (10) find it difficult to focus on one project for a long time | **0.357** | | .451 | | **0.299** | 0.354 |
| (5) not comfortable trying new ways of doing things | **0.250** | | .251 | | **0.133** | 0.272 |
| (8) I set goals, but after a while I decide on a new set of goals | -0.013 | | .199 | | **0.231** | -0.035 |

Note: Loadings in bold indicate significant loading onto the factor.

Our 1-factor solution demonstrated sub-optimal model fit, ($\chi2$ = 657.997, CFI = .809, TLI = .767, RMSEA = .104, AIC = 32904.41). In our 2-factor solution, Persistence and Self-composure exhibited significant covariation (.166) and improved fit based on Hu & Bentler (1995) indices [39] cut-offs (AIC = 30528.80). The Chi-square ($\chi^2$) was statistically significant ($\chi^2$ = 378.4, $p < .001$). The Root Mean Square Error of Approximation (RMSEA) was .074, indicating a reasonable error of approximation. Analysis of model fit revealed that both the CFI (.918) and the TLI (.875) indicate an adequate fit [40]. Typically, a significant Chi-square ($\chi^2$) can indicate a lack of model fit [41]. However, $\chi^2$ is affected by sample size and because CFA typically utilises a large sample, reporting a statistically significant chi-square is relatively common [41]. Models whose RMSEA is 0.10 or more, have a poor fit, while a value of .08 or less indicates a reasonable model fit [42,43]. For both the Comparative Fit Index (CFI) and the Tucker-Lewis coefficient (TLI), values close to 1 indicate a very good fit [44,45]. Therefore, the model is an acceptable fit to the sample data based on commonly accepted thresholds ($\chi2$ = 378.4, df = 53, $p < .01$, CFI = .92, TLI = .88, RMSEA = .07). Thus, it can be concluded that the two latent factors (persistence and self-composure) are relatively strong reflections of the associated observed variables and the two-factor model fits the data quite well [39]. Yet, Persistence and Self-composure are hypothesised to be sub-scales of Tenacity. As such, a bifactor CFA model which can assess for a co-existing general factor alongside specific factors [46] was computed. Items loaded on their respective orthogonal specific factors (Persistence and Self-Composure) and a general factor (Tenacity). This model displayed the best fit indices ($\chi2$ = 229.30, CFI = .93, TLI = .89, RMSEA = .07, AIC = 30377.09) and supports a general factor of Tenacity.

## Reliability and validity

Internal consistency estimates of $> .80$ were sought [47]. Analysis revealed that the internal consistency of BUSS is good, reporting a Cronbach's coefficient alpha for the total BUSS score of .8 and McDonald's omega of .79. Alpha and omega reliability estimates for Persistence, .83 and .84 respectively also demonstrate good internal consistency. Similar to the original psychometric testing of BUSS, the Self-composure factor appeared to show lower levels of internal consistency reliability ($\alpha = .57$, $\omega = .58$). These values are similar to those reported by in previous research [1].

Table 4 demonstrates that BUSS positively correlates with the other measures used in this study. Total BUSS and both factors positively correlate with self-control, future work self, resilience, mental well-being and grit. This demonstrates good convergent validity of the BUSS.

**Table 4. Correlation between persistence, self-composure and other measures used.**

| Measure Taken | 1 | 2 | 3 | 4 | 5 | 6 | 7 | 8 |
|---|---|---|---|---|---|---|---|---|
| 1. Persistence | - | | | | | | | |
| 2.Self-Composure | .326** | - | | | | | | |
| 3. Self-Control | .512** | .607** | - | | | | | |
| 4. Future Work Self | .570** | .253** | .315** | - | | | | |
| 5. Resilience | .730** | .340** | .424** | .472** | - | | | |
| 6. Mental Well-being | .642** | .300** | .384** | .524** | .642** | - | | |
| 7. Grit | .606** | .615** | .642** | .437** | .500* | .469** | - | |
| 8. Total BUSS | .884** | .729** | .661** | .529** | .690** | .603** | .733** | - |

** Pearson Correlation is significant at the 0.01 level (2-tailed).

**Table 5. Results from a multiple linear regression for predicting tenacity (BUSS).**

|  | B (unstandardized) | SE B | B (Standardised) | t | p | 95% CI |
|---|---|---|---|---|---|---|
| Constant | 7.015 | 6.77 |  | 10.36 | < .001 | 5.69; 8.34 |
| Self-control | .259 | .020 | .284 | 12.91 | < .001 | .220; .299 |
| Future work self | .155 | .024 | .129 | 6.32 | < .001 | .107; .203 |
| Resilience | .282 | .022 | .292 | 12.57 | < .001 | .238; .325 |
| Well-being | .071 | .016 | .101 | 4.31 | < .001 | .039; .103 |
| Grit | .371 | .028 | .312 | 13.20 | < .001 | .316; .426 |

Note: R2 adjusted = .747; 95%; SE B = standard error for the unstandardized beta; t = independent samples t-test score; CI = confidence interval for B [Lower Bound; Upper Bound].

A multiple linear regression was performed to predict tenacity (DV) based on self-control, future work self, resilience, well-being and grit as predictor variables (see Table 5). Multicollinearity estimates were within the normal range and the homoscedasticity assumption was not violated. A significant regression equation was found (F(5,884) = 525.34, p < .001), with an $R^2$ of .748. Findings indicate that 74.8% of the variation in tenacity can be accounted for by self-control, future work self, resilience, well-being and grit. Participants' predicted tenacity is equal to 7.015 + .259 (SELF-CONTROL) + .155 (FUTURE WORK SELF) + 0.282 (RESILIENCE) + .071 (WELL-BEING) + .371 (GRIT), where all were coded as a total score. Participants' tenacity increased as self-control, future work self, resilience well-being and grit increased. Therefore, self-control, future work self, resilience, well-being and grit are all significant predictors of tenacity (BUSS).

## Discussion

### Main findings

The present study allowed for the examination of a new psychometric measure, the BUSS, to measure tenacity in university students on an international level. Analysis showed that the factor structure was replicated from previous research [1] with all factor loadings reported as statistically meaningful. Confirmatory factor analysis suggested a reasonable fit with the data, indicating that the two-factor model fits the data quite well. However, a bifactor model supports a general factor of Tenacity. This study utilised a large sample size to ensure factor stability [48,49]. For instance, ten participants to each item is widely considered the recommendation to ensure factor stability [50], which would require a minimum of 120 participants for a 12-item measure, whereas this study included over 1000 participants. A Cronbach's coefficient alpha of .80 and McDonald's omega of .79 were reported for the total BUSS score, demonstrating good internal consistency reliability. Analysis also found that the BUSS exhibited good convergent validity. Such that, Total BUSS and both factors positively correlated with self-control, future work self, resilience, mental well-being and grit. Table 4 shows that some of these intercorrelations are particularly high, supporting our argument that these individual constructs are strongly related to tenacity and that the combination of these constructs into one, short assessment is highly relevant and useful. Specifically, total BUSS has very high correlations with each of the independent variables (between .529 for future work self and .733 for grit). This can also be said for the factor of persistence, which also has particularly high correlations (between .570 for future work self and .730 for resilience). On the other hand, the factor of self-composure has slightly weaker correlations with other psychological constructs related to tenacity. For instance, its lowest correlation is

with future work self at .253 and its highest with grit at .615. These findings support the factor structure that also demonstrate the self-composure factor of tenacity possessing less relevance and significance. While it does appear to have weaker factor loadings and convergent validity, the self-composure factor still offers a unique and significant contribution towards assessing tenacity.

Finally, a multiple linear regression revealed that self-control, future work self, resilience, well-being and grit were all significant predictors of tenacity (BUSS). Indeed, nearly 75% of the variation in tenacity could be accounted for by self-control, future work self, resilience, well-being and grit (see Table 5). Upon inspection of the unstandardized beta coefficients (see Table 5), it is apparent that self-control, resilience and grit are the strongest of these predictors, which is also backed up by the bivariate correlations shown in Table 4 which show these constructs showing greater correlation with Total BUSS. For instance, for every one unit increase in self-control, the tenacity score increases by .259. The greatest predictor is arguably Grit, with the highest bivariate correlation (.733) and largest unstandardized beta (.371). Although all independent variables are significantly contributing towards the prediction of tenacity. Therefore, BUSS is clearly a distinct construct that is comprised of several relevant and integral components that each contribute significantly and uniquely towards tenacity. Reliability analyses and scale correlations further supported the psychometric properties of the BUSS for international students. Further, this assumes cross validation of the BUSS factor structure and allows for generalisation. Based on the present study, the BUSS can be considered a reliable and valid measure of tenacity for university students internationally, particularly those within Europe, the USA and predominantly English speaking countries.

## Comparisons with previous research

Compared to the original development of the BUSS, which utilised a British sample of university students, this study recruited students internationally from over 25 countries. Despite the diversity of the sample population in this study, psychometric properties of the scale, along with its reliability and validity were supported. When comparing the international factor structure of the BUSS to the original factor structure [1], the same seven items loaded onto factor 1 (persistence). Likewise, the same 5 items loaded onto factor 1 (self-composure). This indicates that the factor structure replicates, when using an international sample of university students. On the other hand, item 8 loaded weakly onto the self-composure factor (.199). The corrected item-total correlation for item 8 was low (.032) and the removal of item 8 led to a small improvement in Cronbach's alpha and McDonald's omega. Taking all things into consideration, item 8 was removed from factor analysis. While item 8 was not considered to be a valuable contributing factor to the self-composure factor, it was retained in the BUSS total score as various indicators pointed out it was a valuable item and contributed towards the measurement of tenacity as a whole. The removal of items could compromise construct coverage for what is arguably only a small increase in internal reliability. The original factor structure was Persistence (7) + Self-composure (5), whereas the international factor structure is Persistence (7) + Self-composure (4) + 1.

As shown in previous research, CFA confirmed two correlated latent constructs of BUSS, persistence and self-composure [1]. However, as supported by our bifactor CFA, the authors recommend modelling tenacity as a general factor making the use of total scores appropriate. The authors considered the possibility that the two latent factors, persistence and self-composure, were an artefact of item wording [51–53]. For instance, the persistence factor is comprised of positively worded items and the self-composure factor is made up of negatively worded items. Previous research has indicated that the inclusion of positively worded and

negatively worded items in a questionnaire can result in artefacts which impact on the number of factors resulting form factor analysis [53,54]. Moreover, there are several wording effects that are associated with responses to negatively worded items, such as increased cognitive load as a consequence of switching between response formats and the presence of socially desirable answers [55]. Nonetheless, the authors believe that the two specific factors reflects conceptually distinct sub-constructs, persistence and self-composure. We are most concerned with tenacity as a total measure, and the bifactor CFA analysis indicated that the two factors combined into a general factor is more coherent, meaningful and predictive. Similarly, following the extraction of a two-factor structure model of grit, Duckworth et al., (2007) proposed the use of total grit score alone [2]. Therefore, we recommend utilising BUSS as a general factor (unidimensional) measure of tenacity.

## Limitations, implications and recommendations for future research

This study provides a generalised international tool, the BUSS, to measure tenacity in university students across the world. Nevertheless, further research is required to continue the investigation into the cultural differences of tenacity and what characteristics contribute towards academic success and well-being in university students around the globe? As mentioned, these characteristics are thought to be universal and present in university students around the world. However, it remains unclear if the construct of persistence is equivalent across different countries and cultures. Due to some underrepresentation from certain countries in this study, it is not possible to conduct tests of invariance for all of the different countries included in the sample. Thus, future research is needed to conduct a measurement invariance study across these countries to further explore this. The generalizability of current study is limited due to the underrepresentation of some countries on Prolific. As some countries have less of a presence on Prolific, there is greater participation from countries where use of the site is more popular [23]. In addition, it is unclear to what extent the sample of students recruited via Prolific are representative of a general international student population. Both of these things together mean that the sample of participating students may not be representative of an international student population. The authors welcome further investigation and international comparisons.

Our CFA have made use of commonly used cut-offs for fit indices [39]. These fit indices are derived from simulation studies and appropriate to use under the conditions in which they were derived. As our models do not reflect the same conditions by which those fit indices were derived, our use of arbitrary cut-offs may make precise assessment of model fit or mis-specification difficult. The use of dynamic fit indices for CFA [56] is a novel development which would improve our ability to discern good model fit. At present, however, dynamic fit indices for bifactor models cannot be obtained. To avoid confusion and misrepresentation, we have not included dynamic fit indices for the 1- and 2-factor models as these would be judged by a different standard than the bifactor model. We recommend further validation efforts using dynamic fit indices, and remain tentative with regards to model fit whilst using the arbitrary fit indices cut-offs.

Further, this study did not assess the criterion validity of BUSS in that it does not evaluate the extent to which tenacity (BUSS) is related to students success, satisfaction with their academic performance or their intention to drop out. It can only be assumed that because the BUSS is comprised of a multitude of concepts that are known to be strongly relevant for academic outcomes, that the BUSS too will be a useful tool to predict such outcomes. However, further analyses are needed to confirm the importance of BUSS for educational success and academic outcomes.

Users of Prolific receive payment for participating in online research, and participants of this study received £1.60 for 10 minutes of their time which equates to £9.60 per hour. Once considered an unsuitable method of recruitment, offering financial gain for participating in research is increasingly becoming common practice [57] and is argued to be an ethically acceptable method of recruitment [58]. The self-reporting nature of this study also raises the possibility of social desirability much like with most quantitative studies [59].

Nevertheless, it can be argued that the use of BUSS could provide a shortcut to assess an array of highly relevant and important psychological constructs in one time efficient and economical solution. BUSS can help higher education institutions, academics and educators to better understand their student population. By gaining knowledge of their students' tenacity, this allows educators the opportunity to support their students and help to guide at-risk students towards targeted positive psychology education programs. We also suggest that the BUSS be used to further investigate the role that tenacity plays on various academic, physical, psychological, social and life-long outcomes. Further research should explore the extent to which encouraging tenacity at university can influence future life prospects.

## Supporting information

**S1 File.**
(SAV)

## Acknowledgments

The authors would like to acknowledge the support provided by Professor Patrick McGhee and Dr. Gill Waugh.

## Author Contributions

**Conceptualization:** Chathurika Kannangara, Jerome Carson.

**Data curation:** Chathurika Kannangara, Rosie Allen, Jerome Carson.

**Formal analysis:** Rosie Allen, Kevin D. Hochard.

**Funding acquisition:** Chathurika Kannangara.

**Investigation:** Chathurika Kannangara, Rosie Allen, Kevin D. Hochard, Jerome Carson.

**Methodology:** Chathurika Kannangara, Rosie Allen, Jerome Carson.

**Project administration:** Rosie Allen.

**Software:** Rosie Allen.

**Supervision:** Chathurika Kannangara, Jerome Carson.

**Validation:** Rosie Allen, Kevin D. Hochard.

**Visualization:** Chathurika Kannangara, Jerome Carson.

**Writing – original draft:** Rosie Allen.

**Writing – review & editing:** Chathurika Kannangara, Kevin D. Hochard, Jerome Carson.

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
