## [Decision Letter · Decision Letter 0]

23 Nov 2021

PONE-D-21-30746An International Validation of the Bolton Unistride Scale (BUSS) of Academic Tenacity.PLOS ONE

Dear Dr. Allen,

Thank you for submitting your manuscript to PLOS ONE. After careful consideration, we feel that it has merit but does not fully meet PLOS ONE’s publication criteria as it currently stands. Therefore, we invite you to submit a revised version of the manuscript that addresses the points raised during the review process.

We look forward to receiving your revised manuscript.

Kind regards,

Frantisek Sudzina

Academic Editor

PLOS ONE

Journal Requirements:

- https://journals.plos.org/plosone/article/file?id=10.1371%2Fjournal.pone.0235157&type=printable

The text that needs to be addressed involves the results, specifically one of the tables.

In your revision ensure you cite all your sources (including your own works), and quote or rephrase any duplicated text outside the methods section. Further consideration is dependent on these concerns being addressed.

Reviewers' comments:

Reviewer's Responses to Questions

**Comments to the Author**

1. Is the manuscript technically sound, and do the data support the conclusions?

Reviewer #1: Yes

Reviewer #2: Partly

2. Has the statistical analysis been performed appropriately and rigorously? 

Reviewer #1: Yes

Reviewer #2: No

3. Have the authors made all data underlying the findings in their manuscript fully available?

Reviewer #1: Yes

Reviewer #2: No

4. Is the manuscript presented in an intelligible fashion and written in standard English?

Reviewer #1: Yes

Reviewer #2: Yes

5. Review Comments to the Author

Reviewer #1: This is an important paper as it provides evidence that the BUSS may be applied in international settings; the BUSS being a novel measure of personal characteristics associated with academic tenacity. I enjoyed reading the paper; it is well written and presented in clear and logical fashion; entirely in keeping with the reporting of high quality psychometric research. I have no criticism of the statistical approach. I am of the opinion that this paper represents a significant contribution to the assessment of student characteristics and provides a very sound basis for further research.

Reviewer #2: I was having a hard time reviewing this paper and coming to a conclusion: on the one hand, there are several aspects that I liked in the paper. I think that the overall idea of the BUSS is a good and important one and that the collected sample is big and the confirmatory approach is recommendable. However, on the other hand, I see some major issues in the analyses (some actually deal-breaking) and the reasoning of what the BUSS can do and why it should be used instead of other scales. In this review, I strongly focus on the methodological aspects and the problems with the analysis. This is not because the other parts are already completely fine (other reviewers may raise important points there). Instead, I think that depending on the changes in the analyses and results substantial rewriting may be necessary for the introduction and discussion as well. If there are any points unclear in the following review (see attachment), please do not hesitate to contact me directly (I sign my reviews openly as I think that scientific communication is important, and the ability to contact a reviewer when something is not clear outweighs the benefits of ‘blind’ review).

Tom Scherndl (thomas.scherndl@plus.ac.at)

6. PLOS authors have the option to publish the peer review history of their article (what does this mean?). If published, this will include your full peer review and any attached files.

Reviewer #1: No

Reviewer #2: **Yes: **Thomas Scherndl

---

## [Author Response · Author response to Decision Letter 0]

17 Feb 2022

Journal Requirements:

*Thank you, these have been amended as per your requirements.

- https://journals.plos.org/plosone/article/file?id=10.1371%2Fjournal.pone.0235157&type=printable

The text that needs to be addressed involves the results, specifically one of the tables.

In your revision ensure you cite all your sources (including your own works), and quote or rephrase any duplicated text outside the methods section. Further consideration is dependent on these concerns being addressed.

*Unfortunately, I can’t see where in the results the text or tables presented overlaps with the previous publication?

*Apologies for this, we have uploaded the anonymised data file as supporting information with the submission.

*Please see above comment. We would greatly appreciate you amending our data availability statement to reflect these changes.

*References have been corrected. Kannangara et al 2018 was kept in the references as it is now cited in Table 2 to show where Sample 1 and 3 came from. Wang & Gordon (2012) was changed to the correct reference.

5. Review Comments to the Author

Reviewer #1: This is an important paper as it provides evidence that the BUSS may be applied in international settings; the BUSS being a novel measure of personal characteristics associated with academic tenacity. I enjoyed reading the paper; it is well written and presented in clear and logical fashion; entirely in keeping with the reporting of high quality psychometric research. I have no criticism of the statistical approach. I am of the opinion that this paper represents a significant contribution to the assessment of student characteristics and provides a very sound basis for further research.

*Thank you very much for your comments on our manuscript. We are delighted that you support our research and enjoyed reading our manuscript. It is greatly appreciated.

Reviewer #2: I was having a hard time reviewing this paper and coming to a conclusion: on the one hand, there are several aspects that I liked in the paper. I think that the overall idea of the BUSS is a good and important one and that the collected sample is big and the confirmatory approach is recommendable. However, on the other hand, I see some major issues in the analyses (some actually deal-breaking) and the reasoning of what the BUSS can do and why it should be used instead of other scales. In this review, I strongly focus on the methodological aspects and the problems with the analysis. This is not because the other parts are already completely fine (other reviewers may raise important points there). Instead, I think that depending on the changes in the analyses and results substantial rewriting may be necessary for the introduction and discussion as well. If there are any points unclear in the following review (see attachment), please do not hesitate to contact me directly (I sign my reviews openly as I think that scientific communication is important, and the ability to contact a reviewer when something is not clear outweighs the benefits of ‘blind’ review).

Tom Scherndl (thomas.scherndl@plus.ac.at)

*We would like to thank you very much for your detailed and incredibly useful comments on our manuscript. We really appreciate the quality of your review and truly believe that the paper is better after making your recommended revisions. 

2021-11-17

Review for ‘An International Validation of the Bolton Unistride Scale (BUSS) of Academic‘

I was having a hard time reviewing this paper and coming to a conclusion: on the one hand, there are several aspects that I liked in the paper. I think that the overall idea of the BUSS is a good and important one and that the collected sample is big and the confirmatory approach is recommendable. However, on the other hand, I see some major issues in the analyses (some actually deal-breaking) and the reasoning of what the BUSS can do and why it should be used instead of other scales. In this review, I strongly focus on the methodological aspects and the problems with the analysis. This is not because the other parts are already completely fine (other reviewers may raise important points there). Instead, I think that depending on the changes in the analyses and results substantial rewriting may be necessary for the introduction and discussion as well. If there are any points unclear in the following review, please do not hesitate to contact me directly (I sign my reviews openly as I think that scientific communication is important, and the ability to contact a reviewer when something is not clear outweighs the benefits of ‘blind’ review). 

Tom Scherndl (thomas.scherndl@plus.ac.at) 

General

• The authors named the first factor of the scale ‘tenacity’ but also seem to use this as the name for the total BUSS score and the overall construct: this was confusing for me – a simple fix would be to be more precise in the wording?

*We have changed the name of the “tenacity” factor to “persistence” and ensures this is coherent throughout the manuscript. We have therefore also decided to change “academic tenacity” to “tenacity”.

• Table 2: Samples used: it was not always completely clear to me as a casual reader where those samples in Table 2 came from and what the relevance was… are any of those samples published? Are they newly collected for this study? Additionally, the sample size for the current sample stated in the table does not match with the sample size given in the text (969 vs. 1043). Table 1 suggests that the latter number is correct? Please give more detail why this is relevant for the given paper. Small typo: “On inspection of table 3…” should probably refer to table 2…. 

*we have included references in the Table 2 notes for more information about where the samples came from

*the sample size is 1043 as double checked in Table 2 and the datafile. This has been amended in Table 2 (p.7). Thank you for spotting this mistake. 

*this has been changed to “on inspection of table 2…” (p.6)

• I do not agree with the authors that the minimal factor loading that is ‘recommended’ will/should be .162. It is simply maths when factor loadings will be significant depending on specific sample sizes. You do not need a citation for that. I would also argue that items with such low loadings do not contribute much to a given factor as less than 3% of the given variance in the item is explained by the latent factor. I suggest either completely cutting this part or being more specific about why such a (weak) recommendation is given. The statistical significance of factor loadings is a really weak argument to give when conducting factor analyses. This is also a problem for some items of factor 2 later: as these seem to be consistently loading quite low, you need a good argument why you still keep them as they seem to measure something else…

*we have removed the recommendation that factor loadings should exceed .162 to be retained. We have also discussed corrected item-total correlation, Cronbach’s alpha and McDonald’s omega after removal of items, specifically for items 8 and 5 (which are the two which fall below .4). We have also included mention on how removing these items would impact construct coverage and have therefore decided to retain these items.

Method

• Prolific is a convenient way to sample big samples. However, I miss a critical reflection on the limitations of the resulting sample. Do you think that participants at Prolific are representative of a more general (international) student sample? I am quite skeptical. Typical students in my country are *NOT* users of Prolific at all… perhaps it is different in the US/UK, but you should add some caveat about this potential problem (or give some citations that all is well – I am not an expert with Prolific samples). When looking at your sample you have a very strong emphasis on UK/US/Canada students. Did you check whether they were *actually* from that country based on their IP address or is it only self-report? There was quite a problem with people faking their location on MTURK some years ago (even going so far as to fake their IP address using VPN). How can you be sure that participants were actual students in the mentioned country (and not someone else from a developing country faking being a student to get the money) or even bots? If you have anything to check for quality this may be fine. If not: add something in the limitations that online samples have been prone to such faking. 

*Students self-report their country of study and IP addresses were not checked to support their answers. This has been added into the study limitations as well as your point about the representativeness of the prolific sample.

• You should also add that participants were incentivized (how much?). At least I infer monetary incentivization based on the grant mentioned in the beginning and the fact that everything on Prolific is incentivized. How long did it take participants to take the survey (on average, min/max)? what was the hourly wage: was that a wage that would have encouraged careful responding or just made participants get through quickly? 

*this is an important point that should have been mentioned in the study limitations. Thank you for pointing this out. The offer of financial gain to participate and how much they received has been included (p.20).

• Did you use any quality control measures to avoid careless responding or just ‘clicking through’? This is a lesser problem on Prolific than on MTurk but still, I would strongly suggest checking whether participants were finishing very quickly or giving always the same answers. I do not suppose that you have any attention checks, bogus items or similar measures to make sure that participants read the instructions and questions? Perhaps some reaction/reading times? If yes, you should screen for overly fast responders or participants that failed these attention checks/bogus items. 

*we did in fact include one attention check item and assess the time taken to compelte the survey, as we do with all our online research via platforms such as Prolific. We have included this information in procedure section of methods as we agree that this is important information.

• Description of scales/instruments: this was well done and a nice overview. 

*thank you.

Results

• I was a little bit confused about what was done concerning the factor analysis. The authors say that they did a confirmatory factor analysis and report corresponding fit indices. This is fine and nice (I am strongly in favor of a confirmatory approach): I would however suggest reporting all the fit indices first and then discussing them all in one instead of their piece-meal approach discussing each fit index separately (MINOR). It seems that all point to a similar conclusion thus a more succinct interpretation is ok (but see next bullet point). However, they also mention using PROMAX as a rotation method, as well as Bartlett’s test, etc. All of these things are typically used / more common in EFA contexts which is however not reported (no factor loadings etc.). I have not used AMOS in years, but I find it strange that those indices would be shown in an AMOS output or AMOS uses a promax method in a confirmatory (maximum likelihood?) factor analysis. I have never seen these indices in confirmatory factor analyses (reported in papers as well as in output obtained by R (lavaan) or MPLUS). Is it perhaps a leftover from an exploratory factor analysis or some strange hybrid version that was later dropped from the paper? I would suggest that the paper will be fine if you focus completely on the CFA and drop everything related to the EFA. The EFA does not add anything if there is already enough data and previous papers showing a given factorial structure. So drop the EFA – keep the CFA. 

*the paragraph that presents the fitness of the model has been reorganized to report all the fit indicised and then discussing them. We agree that this now reads better and more clearly, thank you.

* we believe that the focus is on the CFA only and the output from the performed CFA through AMOS v.26 that provides standardized factor loadings, regression weights, intercepts, correlations, covariances, effects, pairwise parameter comparisons and model fit. We are happy to provide the AMOS output file if this is something you would like to see?

*We can confirm we used a maximum likelihood method of factor extraction and have inserted this info in the results section for clarity (p.11).

*Text relating the EFA has been removed to keep the focus on the CFA as recommended. 

• However, as you later-on argue that you prefer to use a composite score instead of the two factors, I would recommend extending the CFA approach: 1) include a CFA model with a single factor and 2) a bifactor model with the 2 specific factors and the overall /general ‘g-factor’ that you are arguing for later in the paper. Compare these 3 models (1 factor, 2 factors, bifactor model) using fit indices and change of CHI² etc. Without having seen your data and only based on the correlations and fit indices: I would expect that the bifactor model will probably have the best fit: this is then a great argument for using the composite score and arguing for a general factor. 

*As suspected by the reviewer, a bifactor model was displayed the best fit indices (χ2=229.302, TLI = .885, RMSEA = .072, AIC= 30377.094), compared the 1-factor (χ2=657.997, TLI = .767, RMSEA = .104, AIC= 32904.413), and the 2-factor solution (χ2=657.997, TLI = .862, RMSEA = .079, AIC= 30528.795). We have incorporated these comparisons in the text to support the argument for a general factor and have added the 1-factor and bifactor coefficients to table 3. 

• Additionally, you may also decide to use the better-suited omega as an estimate for internal consistency instead of Cronbach alpha which has received so much criticism in the last years (rightly so). If you have any problems doing this analysis in SPSS/AMOS (I am not sure if this is possible there – I have not used them for quite some time), you may opt to use JAMOVI as an open-source program with a graphical user interface. For SPSS users, JAMOVI is very easy to use and understand and it includes omega as a reliability estimate out of the box. Obviously R (package psych) would be even better, but I completely understand if you do not want to make the dive into R for this paper. 

*We have added omega as an estimate of internal consistency for the BUSS.

• Fit criteria/cut-offs: Personally, I do not completely agree that the fit of the CFA is fine based on typical cut-offs. However, I am aware that many different cut-offs have been proposed in different papers which may be cited depending on the results. All of those cut-offs are more or less arbitrary (as the specific parameters from those simulations are probably never completely met in a given case) and thus debatable. I would like to raise another possibility to solve this very unsatisfying and taxing debate of whether the proposed criteria are ‘correct’: Other authors have argued that using generic cutoffs are problematic and suggested a new approach(McNeish & Wolf, 2021 in Psychological Methods. https://psyarxiv.com/v8yru for the open-access pre-print). Instead of using cutoffs derived from simulations that may or may not mirror the actual model evaluated in a given study (e.g. using Hu&Bentler or similar recommendations), authors should simply evaluate their obtained fit against the results of a simulation that is based on the model they actually tested. The above-mentioned authors give an easy-to-use app (at least that was my impression when I looked at it after the publication of the article – I have not yet used this approach myself). I would strongly encourage the authors to compare their model fit against those cutoffs derived from simulation. I think that this procedure will (and should) be the recommended approach in the future and that your paper will benefit from it. This will also sidestep the valid criticism that some of your fit indices still hint at (substantial?) misspecification of the model. Maybe this will be less of a problem if you follow my suggestion above to include a bifactor model. It may be that the fit of this other model is so good, that the fit indices are showing good model fit beyond any doubt (but do not count on it). 

*We have made use of the shiny app developed by McNeish & Wolf (2021). We obtain dynamic fit indices for our 1-factor solution; SRMR = .043 to .058, RMSEA = .051 to .09, CFI=.949 to .864 at the 95% level (assuming item residual correlations of .3 are not included for 1/3 to all items in the model). When compared to our actual 1 factor fit indices, SRMR = .072, RMSEA = .104, CFI = .809, we can see this a 1 factor solution is mis-specified relative to simulated data derived cut-offs and suggests modifications to the model are required. Only an SMRM = .024 at the 90% level was provided by the app for our 2-factor model. Our actual SMRM value for our 2-factor model was .061, indicating some degree of misspecification when considering dynamic fit indices. We wished to assess our bifactor model using the dynamic fit indices but the shiny app currently states that bifactor models are “not available yet” (https://www.dynamicfit.app/connect/). On the whole, we agree that a bifactor model is likely most suitable but must remain cautious as we are presently only able to asses its fit using the arbitrary Hu & Bentler fit indices cut-offs. 

As we are unable to obtain dynamic cut-offs for all models, we are concerned that the use of dynamic cut-off for some models and not others could lead to mis-interpretations or confusion. We have therefore retained the Hu & Bentler cut-offs in our results section but added to the discussion that dynamic estimates should be considered and are worth further research, once dynamic fit indices can be computed for bifactor models. 

• Based on the reported fit, I would suspect that there are substantial residual error correlations hidden in the residuals – you may want to check the modification indices. Be very careful when applying any of those: however, sometimes they do make a lot of sense and can be argued on theoretical grounds as well. If you do a bifactor model as mentioned above, I would expect that those modification indices are becoming less relevant – overall suggesting better model fit. 

*Modification indices were checked for the 1-factor and 2-factor models but AIC of the bifactor model remained superior. We have left these out so as to not confuse reader without clear theoretical underpinning for the use of additional residual error correlations in our models.

• Results concerning discriminant and convergent validity: I have two concerns in this area: 

1) the correlations between BUSS and all the other scales are consistently high: if I may raise the question: is it too high? It suggests substantial overlap with GRIT, resilience, etc.– what did you expect? Is having a scale that is highly correlated with other (sometimes more established, always more specific) scales really better? Why should we prefer the BUSS over the other scales when measuring relevant psychological constructs? 

*we have included some discussion on the matter on convergent validity. We agree that this is an important point we don’t feel the correlations are ‘too high’. It supports our argument that tenacity is comprised of several highly relevant and related constructs that each offer a unique and significant contribution towards BUSS. BUSS allows the assessment of all of these important and relevant constructs, that are expected to be highly related, in a timely and reliable solution.

2) Discriminant validity: I have a strong objection here: the authors *do not* check discriminant validity and use the term in a faulty way. Discriminant validity is the degree to which scores on a test *do not* correlate with scores from other tests that *are not* designed to assess the same construct. The artificial dichotomization and comparing the extreme groups does not add anything at all to the question of whether BUSS has discriminant validity. The BUSS and other scales are highly correlated: it is obvious that if you cut the BUSS into extreme groups, these will differ in the other scales – it is just rephrasing the correlation in another analysis. As there is also no real theoretical argument why the mentioned cuts are used (I suppose the first and last quartile are contrasted – but why are these cutoffs even interesting/meaningful?), I would suggest dropping this group comparison completely and sticking with the correlations. 

*analysis that used the cut off points of high and low BUSS were removed from the results and mention of these analyses were removed from the rest of the manuscript. 

• The report of the multiple regression has to be more complete: It is very uncommon in psychological assessment papers to just report a regression formula. I am not even sure what was reported in the formula: standardized betas or unstandardized b? I would strongly recommend including a complete table comprising standardized as well as unstandardized parameters and respective confidence intervals to better judge the worth of the individual predictors. It would be of interest to compare the bivariate correlations to the respective betas in the discussion: which predictors actually have a unique contribution and what is partialled out in the regression? This would strengthen the point of the paper that the BUSS is on the one hand something distinct, but also a combination of relevant other scales and constructs. In other words: can the BUSS be used as a shortcut to assess a multitude of relevant psychological aspects in a very economic way? I think this may be a viable argument to make for the authors: however, I did not find that as the main point of the reasoning. Perhaps I am wrong – I am open to other arguments as well, although right now, I do not get any good points on why this scale can/should be used in contrast to the other scales. 

*a table (Table 5) has been included to show the standardized and unstandardized betas, along with confidence intervals (p.17).

*comparisons have been made between the bivariate correlations (Table 4) and the beta coeffiecients (Table 5) which do in fact help to make a case for the relevance and convenience of BUSS. Many thanks for this helpful suggestion. (p.19 and p.22).

Discussion/Limitations

• Summary of the results is mostly ok concerning the factorial structure, although overreaching for the validity claims (see above regarding discriminant validity, but also what is ‘good’ convergent validity: too high an overlap can actually be a problem if the other scales should measure distinct other constructs).

o I do not concur with the idea that n = 120 is enough to ensure factor stability of a scale of 10 items and think that this rule of thumb is ridiculous. However, I will not debate that much. I still think it is wrong and you just could cut this part. I suppose it was inserted in a previous round of reviews, so I will not force you to delete something that was probably forced on you. 

o ‘can be utilised in populations across the world’ – you should insert ‘student’ here to be more precise… I think this is overreaching … you have a sample comprised of mostly English-speaking countries with some additional European countries. 

o ‘strong relationship between academic tenacity and the two comprising factors was observed’: not sure what you mean. If you mean that the overall BUSS score is correlated with the two factors…. Well, this is because the total score includes the two factors… if this is what you meant: please cut it, as this is obvious and not an argument for the BUSS at all. 

*we have included more discussion surrounding the convergent validity of the BUSS (p.18-19).

*this has been changed to read “…the BUSS can be considered a reliable and valid measure of tenacity for university students internationally, , particularly those within predominantly English speaking countries.” (p.20)

*we agree and this sentence has been removed (p.18).

• I strongly missed the severe limitation that there was no test at all concerning criterion validity: the authors did not show that BUSS was related to students’ success (e.g. GPA), satisfaction with their academic performance, or their intention to drop out. The BUSS was only correlated with other scales which are more or less theoretically distinct from the BUSS and each other. However, this is not sufficient and persuasive for including and using the BUSS in a study that aims at helping students in their academic life (as suggested). I am aware that you probably do not have the necessary data to investigate criterion validity in this sample. I strongly encourage you to include something like that in upcoming studies and use a more cautious tone in this paper about validity. 

*you are right in saying we did not have access to this data and we completely agree that this is a major limitation of the study (p.22) , considering we are arguing that the BUSS will help assess academic outcomes. We have pointed this out in the limitations and are actually in the midst of a study that investigates this. 

• I do not see where the sweeping recommendations regarding the BUSS are coming from. I think there is still quite some work to do to establish the BUSS psychometrically (especially criterion validity, measurement invariance when comparing different countries, etc.). 

*we have revised the recommendations section of the manuscript to appear less bold or presumptuous (p.23).

General points

• I would strongly encourage the authors to adopt open science practices by making their code as well as their data public to increase transparency and reproducibility. Uploading the data and syntax to e.g. OSF is easy and will help readers to better judge the robustness and impact of the given results. 

*the datafile has now been submitted as supporting information alongside the revised manuscript.

---

## [Editor Report · Decision Letter 1]

21 Feb 2022

An International Validation of the Bolton Unistride Scale (BUSS) of Tenacity.

PONE-D-21-30746R1

Dear Dr. Allen,

We’re pleased to inform you that your manuscript has been judged scientifically suitable for publication and will be formally accepted for publication once it meets all outstanding technical requirements.

Kind regards,

Frantisek Sudzina

Academic Editor

PLOS ONE
---

## [Editor Report · Acceptance letter]

4 Mar 2022

PONE-D-21-30746R1 

An International Validation of the Bolton Unistride Scale (BUSS) of Tenacity. 

Dear Dr. Allen:

I'm pleased to inform you that your manuscript has been deemed suitable for publication in PLOS ONE. Congratulations! Your manuscript is now with our production department. 

Kind regards, 

on behalf of

Dr. Frantisek Sudzina 

Academic Editor

PLOS ONE